# Computational Studies on Diverse Characterizations of Molecular Descriptors for Graphyne Nanoribbon Structures

**DOI:** 10.3390/molecules28186597

**Published:** 2023-09-13

**Authors:** Muhammad Awais Raza, Muhammad Khalid Mahmood, Muhammad Imran, Fairouz Tchier, Daud Ahmad, Muhammad Kashif Masood

**Affiliations:** 1Department of Mathematics, University of the Punjab, Lahore 54590, Pakistan; awaisraza.math@gmail.com (M.A.R.); khalid.math@pu.edu.pk (M.K.M.); daud.math@pu.edu.pk (D.A.); 2Department of Mathematical Sciences, United Arab Emirates University, Al Ain P. O. Box 15551, United Arab Emirates; 3Mathematics Department, King Saudi University, Riyadh 145111, Saudi Arabia; ftchier@ksu.edu.sa; 4Hebei Advanced Thin Film Laboratory, College of Physics, Hebei Normal University, Shijiazhuang 050024, China; malikkashif232@gmail.com

**Keywords:** pharmaceutical materials, molecular descriptors, distance-degree-based molecular descriptors, molecular symmetry, graphyne nanoribbon structures

## Abstract

Materials made of graphyne, graphyne oxide, and graphyne quantum dots have drawn a lot of interest due to their potential uses in medicinal nanotechnology. Their remarkable physical, chemical, and mechanical qualities, which make them very desirable for a variety of prospective purposes in this area, are mostly to blame for this. In the subject of mathematical chemistry, molecular topology deals with the algebraic characterization of molecules. Molecular descriptors can examine a compound’s properties and describe its molecular topology. By evaluating these indices, researchers can predict a molecule’s behavior including its reactivity, solubility, and toxicity. Amidst the captivating realm of carbon allotropes, γ-graphyne has emerged as a mesmerizing tool, with exquisite attention due to its extraordinary electronic, optical, and mechanical attributes. Research into its possible applications across numerous scientific and technological fields has increased due to this motivated attention. The exploration of molecular descriptors for characterizing γ-graphyne is very attractive. As a result, it is crucial to investigate and predict γ-graphyne’s molecular topology in order to comprehend its physicochemical characteristics fully. In this regard, various characterizations of γ-graphyne and zigzag γ-graphyne nanoribbons, by computing and comparing distance-degree-based topological indices, leap Zagreb indices, hyper leap Zagreb indices, leap gourava indices, and hyper leap gourava indices, are investigated.

## 1. Introduction

When neighboring carbon atoms undergo hybridization at different levels such as $sp^{3}$, $sp^{2}$, or sp, they give rise to a multitude of allotropes. These allotropes are characterized by the formation of single, double, or even triple bonds between the carbon atoms. Graphite and diamond are two of the most renowned carbon allotropes [1]. The unique pairing and arrangement of different types of bonds in carbon allotropes give rise to their distinctive physical characteristics. These bonds have different lengths, strengths, geometries, and electrical characteristics, which add to the diversity of these allotropes. Graphite is renowned for being opaque and velvety. The creation of unique carbon allotropes has been the subject of extensive investigation, yielding amazing findings. One notable example is fullerene [2], a molecule composed of carbon atoms arranged in a hollow sphere or cage-like structure. Another breakthrough came with the discovery of carbon nanotubes [3], which are cylindrical structures formed by rolling up graphene sheets. Graphene [4], a single layer of carbon atoms arranged in a two-dimensional honeycomb lattice, revolutionized the scientific community due to its exceptional electrical conductivity, mechanical strength, and flexibility. This finding has cleared the door for developments in a variety of disciplines, including energy storage and electronics. Moreover, a chain of biphenylene [5] and carbon [6], also known as carbyne has gained attention as the strongest known material with remarkable properties such as high electrical conductivity and exceptional mechanical strength. Its potential uses in a variety of fields, including nanoelectronics and aerospace engineering are being investigated by researchers.

In the fields of materials science and quantum technology, graphene, “a single layer of carbon atoms”, sparked a revolution. Graphene’s remarkable properties, such as high electrical conductivity [7], exceptional strength [8], and unique electronic behavior opened up new avenues for research and technological advancements. There are strong prospects for use in a variety of fields, including electronics, photonics, catalysis, and energy storage because of graphene’s diverse shapes and characteristics. Two-dimensional (2D) carbon allotropes known as graphynes have emerged as fascinating materials. The carbon atoms of graphynes are organized in a lattice structure similar to that of graphene but with more carbon–carbon bonds. These extra bonds provide graphynes with special electrical and structural characteristics that make them desirable for a variety of applications. The study of graphynes and other 2D materials continues to drive innovation in the field of quantum technology and lays the foundation for the development of future-generation devices and materials.

For over a decade, researchers have been dedicating their efforts to synthesizing a novel form of carbon called graphene, but their attempts have been unsuccessful. However, a team of scientists recently made a remarkable advancement that overcame this obstacle, marking an important turning point in the study of carbon materials. This innovation has resolved a long-standing issue and has led to the successful creation of Carbon’s elusive allotrope [9]. This recently created substance has exceptional qualities and was meticulously designed to rival the conductivity of graphene [10] while providing greater control. This achievement has opened up exciting new avenues for research in the semiconductor, electronics, and optics fields. The discovery has prompted a resurgence of interest and hastened research projects examining the special qualities and prospective uses of this novel carbon allotrope. Given the importance of these networks’ uses, it is critical to conduct research on their molecular topology.

The structure of γ-graphyne [9] is shown in Figure 1.

In this article, we focus on molecular graphs, which are graphical representations of chemical molecules. These molecular graphs consist of vertices that represent atoms and edges that represent the bonds connecting those atoms. Topological indices are quantitative measurements that are derived from the molecular graph’s network and are acquired using an isomorphic graph-invariant method. In studies involving the Quantitative Structure–Property Relationship/Quantitative Structure–Activity Relationship (QSPR/QSAR) [11,12], they found extensive uses and offer insights into the topology of molecular structures. Researchers can glean important knowledge about the connection and arrangement of the molecular structure by using topological indices. These indices serve as quantitative measures that aid in understanding various chemical properties and behaviors of molecules. Overall, in chemical research and studies including QSPR/QSAR, the use of topological indices and other graph invariants provides a potent toolkit for examining the molecular structure and understanding its consequences.

Mathematics, spanning various concepts and tools, plays a crucial role in chemistry by regulating the properties of elements without the need for quantum physics. These mathematical techniques offer insights into molecular features that might not be immediately obvious when combined with an examination of molecular symmetry. Among the various types of topological indices, distance-degree-based indices hold particular significance. These indices are correlated with numerical values that reflect the physical, chemical, and biological aspects of molecules as well as their structural traits. In conclusion, topological indices in chemistry provide an effective mathematical framework for analyzing molecule structures and making a wide range of predictions about their characteristics and behaviors.

The first Zagreb index, denoted as $M_{1}\left(G\right)$, and the second Zagreb index, denoted as $M_{2}\left(G\right)$, were initially introduced by Gutman [13] in 1972. These indices act as crucial graph invariants that provide significant details about the topology and structural characteristics of a particular graph. In a subsequent study by Rafiullah [14], there has been notable research conducted on Zagreb indices for double graphs. Rafiullah’s work expands the use of Zagreb indices to a wider class of graphs, concentrating on double graphs and their distinctive properties. Wiener [15] proposed the first topological index which is distance-based, while exploring paraffin breaking point in 1947, whereas Platt [16] proposed the degree-based topological index while researching alkanes to anticipate their physical properties.

The Zagreb index has served as the basis for the development of various other connectivity indices in graph theory. It has proven to be a flexible method for investigating a variety of molecular characteristics, such as chirality, chemical complexity, ZE-isomerism, and heterosystems, among others. For further information on Zagreb indices and the Platt index, one can refer to [17]. In [18], Awais conducted calculations of sixteen irregularity indices for some benzenoid structures. The degree of irregularity or divergence from a regular structure in the examined benzenoid systems is quantified using these irregularity indices. Nazren et al. [19] provide the exact relationship for the first Zagreb index, second Zagreb index, and hyper Zagreb index, as well as the first Zagreb polynomial and second Zagreb polynomial for several systems. In [20], Gao et al. calculated the exact formula for the Zagreb index and hyper Zagreb index of Carbon Nanocones ${\mathrm{CNC}}_{k}\left[n\right]$ and defined the second hyper Zagreb index, which was a degree-based topological index.

In [21], Ghani characterizes some chemical network entropies by using *k*-banhatii topological indices. In [22], Jiao Shi has worked on graphene sheets. In [23], Gutman listed 26 topological indices. In the same publication, he introduced a sombor index. In [24], Dalal Awadh proposed the Gutman connection index of graphs. K. C. Das et al. [25] obtained some mathematical properties, such as lower and upper bonds, as a result of Gutman’s work. Some of the chemical application of the sombor index was obtained in [26]. In [27], Khalid Mahmood worked on the inverse problem for some topological indices.

Naji et al. introduced distance-degree-based topological indices in [28]. Naji et al. proposed the leap Zagreb indices of graph G, which are topological indices based on distance degree in [29]. They observed a substantial relationship between these indices and the physical properties of chemical substances, such as boiling point, DHVAP-standard enthalpy of vaporization, HVAP-enthalpy of vaporization, entropy, and eccentric factor. For more information, please see [30,31]. The proposal of a distance degree has been applied in many other applicable settings. In [32], Sohan Lal has worked on *k*-distance-degree-based topological indices, which include a leap somber index, hyper leap forgotten index, leap Y index, and leap Y coindex.

In [33], Abdul Hakeem calculated some important topological indices for γ-graphyne. In this paper, we compute distance-degree-based topological indices, the leap Zagreb indices, hyper leap Zagreb indices, leap gourava indices, and hyper leap gourava indices to characterize γ-graphyne and zigzag graphyne nanoribbon. Furthermore, their numerical computations and verifications are carried out.

## 2. Basic Aspects of Molecular Descriptors

In this section, we focused on some essential and novel concepts connected to *k*-distance-degree-based topological indices.

Consider a molecular graph Γ with V(Γ) as the vertex set and E(Γ) as the edge set. The degree of a vertex is the number of vertices adjacent to it that have an edge. The *k*-neighborhood of a vertex $\delta_{1}$ is defined as $N_{k}\left(\delta_{1}\right)=\left\{\delta_{2}\in V\left(\Gamma \right):d\left(\delta_{1},\delta_{2}\right)=k,k\in N\right\}$, where $d(\delta_{1},\delta_{2})$ denotes the shortest path joining $\delta_{1}$ and $\delta_{2}$. The *k*-degree of a vertex $\delta_{1}$ in Γ is the number of *k*-neighbors of vertex $\delta_{1}$ and is denoted by $deg_{k}\left(\delta_{1}\right)$. In this paper, we compute topological indices based on *k*-distance degree for *k* = 2.

V.R. Kulli [34] presented the first leap Zagreb index, which is defined as:(1)$$\begin{array}{c} LM_{1}\left(\Gamma \right)=\sum_{\delta_{1}\delta_{2}\in E\left(\Gamma \right)}\left[deg_{2}\left(\delta_{1}\right)+deg_{2}\left(\delta_{2}\right)\right] \end{array}$$

Naji et al. [29] presented the second leap Zagreb index, which is defined as:(2)$$\begin{array}{c} LM_{2}\left(\Gamma \right)=\sum_{\delta_{1}\delta_{2}\in E\left(\Gamma \right)}\left[deg_{2}\left(\delta_{1}\right)\cdot deg_{2}\left(\delta_{2}\right)\right] \end{array}$$

V.R. Kulli [34] presented the hyper leap Zagreb indices, which are defined as:(3)$$\begin{array}{ccc} HLM_{1}\left(\Gamma \right) & = & \sum_{\delta_{1}\delta_{2}\in E\left(\Gamma \right)}{\left[deg_{2}\left(\delta_{1}\right)+deg_{2}\left(\delta_{2}\right)\right]}^{2} \end{array}$$(4)$$\begin{array}{ccc} HLM_{2}\left(\Gamma \right) & = & \sum_{\delta_{1}\delta_{2}\in E\left(\Gamma \right)}{\left[deg_{2}\left(\delta_{1}\right)\cdot deg_{2}\left(\delta_{2}\right)\right]}^{2} \end{array}$$

V.R. Kulli [35] presented the leap gourava and hyper leap gourava indices, which are defined as:(5)$$\begin{array}{ccc} LGO_{1}\left(\Gamma \right) & = & \sum_{\delta_{1}\delta_{2}\in E\left(\Gamma \right)}\left[\left(deg_{2}\left(\delta_{1}\right)+deg_{2}\left(\delta_{2}\right)\right)+\left(deg_{2}\left(\delta_{1}\right)\cdot deg_{2}\left(\delta_{2}\right)\right)\right] \end{array}$$(6)$$\begin{array}{ccc} LGO_{2}\left(\Gamma \right) & = & \sum_{\delta_{1}\delta_{2}\in E\left(\Gamma \right)}\left[\left(deg_{2}\left(\delta_{1}\right)+deg_{2}\left(\delta_{2}\right)\right)\cdot \left(deg_{2}\left(\delta_{1}\right)\cdot deg_{2}\left(\delta_{2}\right)\right)\right] \end{array}$$(7)$$\begin{array}{ccc} HLGO_{1}\left(\Gamma \right) & = & \sum_{\delta_{1}\delta_{2}\in E\left(\Gamma \right)}{\left[\left(deg_{2}\left(\delta_{1}\right)+deg_{2}\left(\delta_{2}\right)\right)+\left(deg_{2}\left(\delta_{1}\right)\cdot deg_{2}\left(\delta_{2}\right)\right)\right]}^{2} \end{array}$$(8)$$\begin{array}{ccc} HLGO_{2}\left(\Gamma \right) & = & \sum_{\delta_{1}\delta_{2}\in E\left(\Gamma \right)}{\left[\left(deg_{2}\left(\delta_{1}\right)+deg_{2}\left(\delta_{2}\right)\right)\cdot \left(deg_{2}\left(\delta_{1}\right)\cdot deg_{2}\left(\delta_{2}\right)\right)\right]}^{2} \end{array}$$

V.R. Kulli [35] presented the sum connectivity leap gourava index and product connectivity leap gourava index, which are defined as:(9)$$\begin{array}{ccc} SLGO(\Gamma ) & = & \sum_{\delta_{1}\delta_{2}\in E\left(\Gamma \right)}\frac{1}{\sqrt{\left[\left(deg_{2}\left(\delta_{1}\right)+deg_{2}\left(\delta_{2}\right)\right)+\left(deg_{2}\left(\delta_{1}\right)\cdot deg_{2}\left(\delta_{2}\right)\right)\right]}} \end{array}$$(10)$$\begin{array}{ccc} PLGO(\Gamma ) & = & \sum_{\delta_{1}\delta_{2}\in E\left(\Gamma \right)}\frac{1}{\sqrt{\left[\left(deg_{2}\left(\delta_{1}\right)+deg_{2}\left(\delta_{2}\right)\right)\cdot \left(deg_{2}\left(\delta_{1}\right)\cdot deg_{2}\left(\delta_{2}\right)\right)\right]}} \end{array}$$

## 3. Molecular Descriptors of γ-Graphyne Nanoribbon Structures

This section contains some results relating to previously defined topological indices based on a two-distance degree, such as the first leap Zagreb index, second leap Zagreb index, first hyper leap Zagreb index, second hyper leap Zagreb index, first leap gourava index, second leap gourava index, first hyper leap gourava index, second hyper leap gourava index, sum connectivity leap gourava index, and product connectivity leap gourava index for γ-graphyne and zigzag graphyne nanoribbon.

### 3.1. γ-Graphyne

In this particular section, we provide an extensive examination of the structure, edge partition technique, and the novel computational results and findings associated with them. Our focus lies on measuring the distance-degree-based topological indices of the γ-graphyne. The detailed structure of γ-graphyne [9,33] is shown in Figure 2 and Figure 3. In each row, the number of vertices is 24mn+12m+12n−6 and the number of edges is 36mn+12m+12n−6. The degree of a vertex $\delta_{1}$ by a distance of two is denoted by $deg_{2}\left(\delta_{1}\right)$, which is the number of vertices that are adjacent to $\delta_{1}$ by the distance of two. The edge partition is shown below:$$\begin{array}{ccc} E_{2,3} & = & \left\{\delta_{1}\delta_{2}\in E\left(\Gamma \right)|deg_{2}\left(\delta_{1}\right)=2,deg_{2}\left(\delta_{2}\right)=3\right\},\;\left|E_{2,3}\right|=8m+8n-4\\ E_{3,5} & = & \left\{\delta_{1}\delta_{2}\in E\left(\Gamma \right)|deg_{2}\left(\delta_{1}\right)=3,deg_{2}\left(\delta_{2}\right)=5\right\},\;\left|E_{3,5}\right|=8m+8n-4\\ E_{5,5} & = & \left\{\delta_{1}\delta_{2}\in E\left(\Gamma \right)|deg_{2}\left(\delta_{1}\right)=5,deg_{2}\left(\delta_{2}\right)=5\right\},\;\left|E_{5,5}\right|=2m+2n+2\\ E_{5,6} & = & \left\{\delta_{1}\delta_{2}\in E\left(\Gamma \right)|deg_{2}\left(\delta_{1}\right)=5,deg_{2}\left(\delta_{2}\right)=6\right\},\;\left|E_{5,6}\right|=12m+12n-12\\ E_{6,6} & = & \left\{\delta_{1}\delta_{2}\in E\left(\Gamma \right)|deg_{2}\left(\delta_{1}\right)=6,deg_{2}\left(\delta_{2}\right)=6\right\},\;\left|E_{6,6}\right|=36mn-18m-18n+12 \end{array}$$

**Theorem** **1.**
*Let ***Γ*** be a molecular graph of γ-graphyne nanoribbon and m,n≥1. Then,*
*(i)* 

$LM_{1}\left(\Gamma \right)=432mn+40m+40n-20$

*(ii)* 

$LM_{2}\left(\Gamma \right)=1296mn-70m-70n+38$

*(iii)* 

$HLM_{1}\left(\Gamma \right)=5184mn-228m-228n+120$

*(iv)* 

$HLM_{2}\left(\Gamma \right)=46656mn-9190m-9190n+4958$




**Proof.** (i)First Leap Zagreb Index$$\begin{array}{ccc} LM_{1}\left(\Gamma \right) & = & \sum_{\delta_{1}\delta_{2}\in E\left(\Gamma \right)}\left[deg_{2}\left(\delta_{1}\right)+deg_{2}\left(\delta_{2}\right)\right]\\ & = & \left(2+3)(8m+8n-4)+(3+5)(8m+8n-4)+(5+5)(2m+2n+2\right)\\ & & +(5+6)(12m+12n-12)+(6+6)(36mn-18m-18n+12)\\ & = & 40m+40n-20+64m+64n-32+20m+20n+20+132m+132n\\ & & -132+432mn-216m-216n+144\\ & = & 432mn+40m+40n-20 \end{array}$$(ii)Second Leap Zagreb Index$$\begin{array}{ccc} LM_{2}\left(\Gamma \right) & = & \sum_{\delta_{1}\delta_{2}\in E\left(\Gamma \right)}\left[deg_{2}\left(\delta_{1}\right)\cdot deg_{2}\left(\delta_{2}\right)\right]\\ & = & \left(2\cdot 3)(8m+8n-4)+(3\cdot 5)(8m+8n-4)+(5\cdot 5)(2m+2n+2\right)\\ & & +(5\cdot 6)(12m+12n-12)+(6\cdot 6)(36mn-18m-18n+12)\\ & = & 48m+48n-24+120m+120n-60+50m+50n+50+360m+360n\\ & & -360+1296mn-648m-648n+432\\ & = & 1296mn-70m-70n+38 \end{array}$$(iii)First Hyper Leap Zagreb Index$$\begin{array}{ccc} HLM_{1}\left(\Gamma \right) & = & \sum_{\delta_{1}\delta_{2}\in E\left(\Gamma \right)}{\left[deg_{2}\left(\delta_{1}\right)+deg_{2}\left(\delta_{2}\right)\right]}^{2}\\ & = & {\left(2+3\right)}^{2}\left(8m+8n-4\right)+{\left(3+5\right)}^{2}\left(8m+8n-4\right)+{\left(5+5\right)}^{2}\left(2m+2n+2\right)\\ & & +{\left(5+6\right)}^{2}\left(12m+12n-12\right)+{\left(6+6\right)}^{2}\left(36mn-18m-18n+12\right)\\ & = & 200m+200n-100+512m+512n-256+200m+200n+200+1452m\\ & & +1452n-1452+5184mn-2592m-2592n+1728\\ & = & 5184mn-228m-228n+120 \end{array}$$(iv)Second Hyper Leap Zagreb Index $$\begin{array}{ccc} HLM_{2}\left(\Gamma \right) & = & \sum_{\delta_{1}\delta_{2}\in E\left(\Gamma \right)}{\left[deg_{2}\left(\delta_{1}\right)\cdot deg_{2}\left(\delta_{2}\right)\right]}^{2}\\ & = & {\left(2\cdot 3\right)}^{2}\left(8m+8n-4\right)+{\left(3\cdot 5\right)}^{2}\left(8m+8n-4\right)+{\left(5\cdot 5\right)}^{2}\left(2m+2n+2\right)\\ & & +{\left(5\cdot 6\right)}^{2}\left(12m+12n-12\right)+{\left(6\cdot 6\right)}^{2}\left(36mn-18m-18n+12\right)\\ & = & 288m+288n-144+1800m+1800n-900+1250m+1250n+1250\\ & & +10800m+10800n-10800+46656mn-23328m-23328n+15552\\ & = & 46656mn-9190m-9190n+4958 \end{array}$$ □

Now, we will calculate leap gourava indices for γ-graphyne nanoribbon.

**Theorem** **2.**
*Let ***Γ*** be a molecular graph of γ-graphyne and m,n≥1. Then,*
*(i)* 

$LGO_{1}\left(\Gamma \right)=1728mn-30\left(m+n\right)+18$

*(ii)* 

$LGO_{2}\left(\Gamma \right)=15552mn-2116\left(m+n\right)+1124$

*(iii)* 

$HLGO_{1}\left(\Gamma \right)=82944mn-13650\left(m+n\right)+7326$

*(iv)* 

$HLGO_{2}\left(\Gamma \right)=6718464mn-1805032\left(m+n\right)+996488$

*(v)* 
*

$SLGO\left(\Gamma \right)=\frac{36}{\sqrt{48}}mn+\left(\frac{8}{\sqrt{11}}+\frac{8}{\sqrt{23}}+\frac{2}{\sqrt{35}}+\frac{12}{\sqrt{41}}-\frac{18}{\sqrt{48}}\right)\left(m+n\right)-\left(\frac{4}{\sqrt{11}}+\frac{4}{\sqrt{23}}-\frac{2}{\sqrt{35}}+\frac{12}{\sqrt{41}}-\frac{12}{\sqrt{48}}\right)$

*
*(vi)* 
*

$PLGO\left(\Gamma \right)=\frac{36}{\sqrt{432}}mn+\left(\frac{8}{\sqrt{30}}+\frac{8}{\sqrt{120}}+\frac{2}{\sqrt{250}}+\frac{12}{\sqrt{330}}-\frac{18}{\sqrt{432}}\right)\left(m+n\right)-\left(\frac{4}{\sqrt{30}}+\frac{4}{\sqrt{120}}-\frac{2}{\sqrt{250}}+\frac{12}{\sqrt{330}}-\frac{12}{\sqrt{432}}\right)$

*



**Proof.** (i)First Leap Gourava Index$$\begin{array}{ccc} LGO_{1}\left(\Gamma \right) & = & \sum_{\delta_{1}\delta_{2}\in E\left(\Gamma \right)}\left[\left(deg_{2}\left(\delta_{1}\right)+deg_{2}\left(\delta_{2}\right)\right)+\left(deg_{2}\left(\delta_{1}\right)\cdot deg_{2}\left(\delta_{2}\right)\right)\right]\\ & = & \left(5+6)(8m+8n-4)+(8+15)(8m+8n-4)+(10+25)(2m+2n+2\right)\\ & & +(11+30)(12m+12n-12)+(12+36)(36mn-18m-18n+12)\\ & = & 88m+88n-44+184m-184n-92+70m+70n+70+492m+492n\\ & & -492+1728mn-864m-864n+576\\ & = & 1728mn-30(m+n)+18 \end{array}$$(ii)Second Leap Gourava Index$$\begin{array}{ccc} LGO_{2}\left(\Gamma \right) & = & \sum_{\delta_{1}\delta_{2}\in E\left(\Gamma \right)}\left[\left(deg_{2}\left(\delta_{1}\right)+deg_{2}\left(\delta_{2}\right)\right)\cdot \left(deg_{2}\left(\delta_{1}\right)\cdot deg_{2}\left(\delta_{2}\right)\right)\right]\\ & = & \left(5\cdot 6)(8m+8n-4)+(8\cdot 15)(8m+8n-4)+(10\cdot 25)(2m+2n+2\right)\\ & & +(11\cdot 30)(12m+12n-12)+(12\cdot 36)(36mn-18m-18n+12)\\ & = & 240m+240n-120+960m+960n-480+500m+500n+500+3960m\\ & & +3960n-3960+15552mn-7776m-7776n+5184\\ & = & 15552mn-2116(m+n)+1124 \end{array}$$(iii)First Hyper Leap Gourava Index$$\begin{array}{ccc} HLGO_{1}\left(\Gamma \right) & = & \sum_{\delta_{1}\delta_{2}\in E\left(\Gamma \right)}{\left[\left(deg_{2}\left(\delta_{1}\right)+deg_{2}\left(\delta_{2}\right)\right)+\left(deg_{2}\left(\delta_{1}\right)\cdot deg_{2}\left(\delta_{2}\right)\right)\right]}^{2}\\ & = & {\left(5+6\right)}^{2}\left(8m+8n-4\right)+{\left(8+15\right)}^{2}\left(8m+8n-4\right)\\ & & +{\left(10+25\right)}^{2}\left(2m+2n+2\right)+{\left(11+30\right)}^{2}\left(12m+12n-12\right)\\ & & +{\left(12+36\right)}^{2}\left(36mn-18m-18n+12\right)\\ & = & 968m+968n-484+4232m+4232n-2116+2450m+2450n+2450\\ & & +20172m+20172n-20172+82944mn-41472m-41472n+27648\\ & = & 82944mn-13650(m+n)+7326 \end{array}$$(iv)Second Hyper Leap Gourava Index$$\begin{array}{ccc} HLGO_{2}\left(\Gamma \right) & = & \sum_{\delta_{1}\delta_{2}\in E\left(\Gamma \right)}{\left[\left(deg_{2}\left(\delta_{1}\right)+deg_{2}\left(\delta_{2}\right)\right)\cdot \left(deg_{2}\left(\delta_{1}\right)\cdot deg_{2}\left(\delta_{2}\right)\right)\right]}^{2}\\ & = & {\left(5\cdot 6\right)}^{2}\left(8m+8n-4\right)+{\left(8\cdot 15\right)}^{2}\left(8m+8n-4\right)+{\left(10\cdot 25\right)}^{2}\left(2m+2n+2\right)\\ & & +{\left(11\cdot 30\right)}^{2}\left(12m+12n-12\right)+{\left(12\cdot 36\right)}^{2}\left(36mn-18m-18n+12\right)\\ & = & 7200m+7200n-3600+115200m+115200n-57600+125000m\\ & & +125000n+125000+1306800m+1306800n-1306800+6718464mn\\ & & -3359232(m+n)+2239488\\ & = & 6718464mn-1805032(m+n)+996488 \end{array}$$(v)Sum Connectivity Leap Gourava Index$$\begin{array}{ccc} SLGO(\Gamma ) & = & \sum_{\delta_{1}\delta_{2}\in E\left(\Gamma \right)}\frac{1}{\sqrt{\left[\left(deg_{2}\left(\delta_{1}\right)+deg_{2}\left(\delta_{2}\right)\right)+\left(deg_{2}\left(\delta_{1}\right)\cdot deg_{2}\left(\delta_{2}\right)\right)\right]}}\\ & = & \frac{1}{\sqrt{5+6}}\left(8m+8n-4\right)+\frac{1}{\sqrt{8+15}}\left(8m+8n-4\right)+\frac{1}{\sqrt{10+25}}\left(2m+2n+2\right)\\ & & +\frac{1}{\sqrt{11+30}}\left(12m+12n-12\right)+\frac{1}{\sqrt{12+36}}\left(36mn-18m-18n+12\right)\\ & = & \frac{36}{\sqrt{48}}mn+\left(\frac{8}{\sqrt{11}}+\frac{8}{\sqrt{23}}+\frac{2}{\sqrt{35}}+\frac{12}{\sqrt{41}}-\frac{18}{\sqrt{48}}\right)\left(m+n\right)\\ & & -\left(\frac{4}{\sqrt{11}}+\frac{4}{\sqrt{23}}-\frac{2}{\sqrt{35}}+\frac{12}{\sqrt{41}}-\frac{12}{\sqrt{48}}\right) \end{array}$$(vi)Product Connectivity Leap Gourava Index$$\begin{array}{ccc} PLGO(\Gamma ) & = & \sum_{\delta_{1}\delta_{2}\in E\left(\Gamma \right)}\frac{1}{\sqrt{\left[\left(deg_{2}\left(\delta_{1}\right)+deg_{2}\left(\delta_{2}\right)\right)\cdot \left(deg_{2}\left(\delta_{1}\right)\cdot deg_{2}\left(\delta_{2}\right)\right)\right]}}\\ & = & \frac{1}{\sqrt{5\cdot 6}}\left(8m+8n-4\right)+\frac{1}{\sqrt{8\cdot 15}}\left(8m+8n-4\right)+\frac{1}{\sqrt{10\cdot 25}}\left(2m+2n+2\right)\\ & & +\frac{1}{\sqrt{11\cdot 30}}\left(12m+12n-12\right)+\frac{1}{12\cdot 36}\left(36mn-18m-18n+12\right)\\ & = & \frac{36}{\sqrt{432}}mn+\left(\frac{8}{\sqrt{30}}+\frac{8}{\sqrt{120}}+\frac{2}{\sqrt{250}}+\frac{12}{\sqrt{330}}-\frac{18}{\sqrt{432}}\right)\left(m+n\right)\\ & & -\left(\frac{4}{\sqrt{30}}+\frac{4}{\sqrt{120}}-\frac{2}{\sqrt{250}}+\frac{12}{\sqrt{330}}-\frac{12}{\sqrt{432}}\right) \end{array}$$ □

### 3.2. Zigzag γ-Graphyne Nanoribbon

In this particular section, we examine the structure, the edge partition approach, and the innovative computational results and insights linked with it in depth. The zigzag γ-graphyne’s distance-degree-based topological indices are the subject of our research. The detailed structure of zigzag γ-graphyne [9,33] is shown in Figure 4. In each row, the number of vertices is 6n+12 and the number of edges is 8n+13. The degree of a vertex $\delta_{1}$ by a distance of two is denoted by $deg_{2}\left(\delta_{1}\right)$, which is the number of vertices that are adjacent to $\delta_{1}$ by the distance of two. The edge partition of the zigzag γ-graphyne for n>2 is shown below:$$\begin{array}{ccc} E_{2,2} & = & \left\{\delta_{1}\delta_{2}\in E\left(\Gamma \right)|deg_{2}\left(\delta_{1}\right)=2,deg_{2}\left(\delta_{2}\right)=2\right\},\;\left|E_{2,2}\right|=2\\ E_{2,3} & = & \left\{\delta_{1}\delta_{2}\in E\left(\Gamma \right)|deg_{2}\left(\delta_{1}\right)=2,deg_{2}\left(\delta_{2}\right)=3\right\},\;\left|E_{2,3}\right|=8\\ E_{3,3} & = & \left\{\delta_{1}\delta_{2}\in E\left(\Gamma \right)|deg_{2}\left(\delta_{1}\right)=3,deg_{2}\left(\delta_{2}\right)=3\right\},\;\left|E_{3,3}\right|=n-2\\ E_{3,5} & = & \left\{\delta_{1}\delta_{2}\in E\left(\Gamma \right)|deg_{2}\left(\delta_{1}\right)=3,deg_{2}\left(\delta_{2}\right)=5\right\},\;\left|E_{3,5}\right|=2n+4\\ E_{5,5} & = & \left\{\delta_{1}\delta_{2}\in E\left(\Gamma \right)|deg_{2}\left(\delta_{1}\right)=5,deg_{2}\left(\delta_{2}\right)=5\right\},\;\left|E_{5,5}\right|=n+4\\ E_{5,6} & = & \left\{\delta_{1}\delta_{2}\in E\left(\Gamma \right)|deg_{2}\left(\delta_{1}\right)=5,deg_{2}\left(\delta_{2}\right)=6\right\},\;\left|E_{5,6}\right|=2n\\ E_{6,6} & = & \left\{\delta_{1}\delta_{2}\in E\left(\Gamma \right)|deg_{2}\left(\delta_{1}\right)=6,deg_{2}\left(\delta_{2}\right)=6\right\},\;\left|E_{6,6}\right|=2n-3 \end{array}$$

**Theorem** **3.**
*Let ***Γ*** be a molecular graph of zigzag γ-graphyne nanoribbon and n>2. Then,*
*(i)* 

$LM_{1}\left(\Gamma \right)=78n+72$

*(ii)* 

$LM_{2}\left(\Gamma \right)=196n+90$

*(iii)* 

$HLM_{1}\left(\Gamma \right)=794n+640$

*(iv)* 

$HLM_{2}\left(\Gamma \right)=5548n-330$




**Proof.** (i)First Leap Zagreb Index$$\begin{array}{ccc} LM_{1}\left(\Gamma \right) & = & \sum_{\delta_{1}\delta_{2}\in E\left(\Gamma \right)}\left[deg_{2}\left(\delta_{1}\right)+deg_{2}\left(\delta_{2}\right)\right]\\ & = & \left(2+2)(2)+(2+3)(8)+(3+3)(n-2)+(3+5)(2n+4)+(5+5)(n+4\right)\\ & & +(5+6)(2n)+(6+6)(2n-3)\\ & = & 8+40+6n-12+16n+32+10n+40+22n+24n-36\\ & = & 78n+72 \end{array}$$(ii)Second Leap Zagreb Index$$\begin{array}{ccc} LM_{2}\left(\Gamma \right) & = & \sum_{\delta_{1}\delta_{2}\in E\left(\Gamma \right)}\left[deg_{2}\left(\delta_{1}\right)\cdot deg_{2}\left(\delta_{2}\right)\right]\\ & = & \left(2\cdot 2)(2)+(2\cdot 3)(8)+(3\cdot 3)(n-2)+(3\cdot 5)(2n+4)+(5\cdot 5)(n+4\right)\\ & & +(5\cdot 6)(2n)+(6\cdot 6)(2n-3)\\ & = & 8+48+9n-18+30n+60+25n+100+60n+72-108\\ & = & 196n+90 \end{array}$$(iii)First Hyper Leap Zagreb Index$$\begin{array}{ccc} HLM_{1}\left(\Gamma \right) & = & \sum_{\delta_{1}\delta_{2}\in E\left(\Gamma \right)}{\left[deg_{2}\left(\delta_{1}\right)+deg_{2}\left(\delta_{2}\right)\right]}^{2}\\ & = & {\left(2+2\right)}^{2}\left(2\right)+{\left(2+3\right)}^{2}\left(8\right)+{\left(3+3\right)}^{2}\left(n-2\right)+{\left(3+5\right)}^{2}\left(2n+4\right)\\ & & +{\left(5+5\right)}^{2}\left(n+4\right)+{\left(5+6\right)}^{2}\left(2n\right)+{\left(6+6\right)}^{2}\left(2n-3\right)\\ & = & 32+200+36n-72+128n+512+100n+400+242n+288n-432\\ & = & 794n+640 \end{array}$$(iv)Second Hyper Leap Zagreb Index$$\begin{array}{ccc} HLM_{2}\left(\Gamma \right) & = & \sum_{\delta_{1}\delta_{2}\in E\left(\Gamma \right)}{\left[deg_{2}\left(\delta_{1}\right)\cdot deg_{2}\left(\delta_{2}\right)\right]}^{2}\\ & = & {\left(2\cdot 2\right)}^{2}\left(2\right)+{\left(2\cdot 3\right)}^{2}\left(8\right)+{\left(3\cdot 3\right)}^{2}\left(n-2\right)+{\left(3\cdot 5\right)}^{2}\left(2n+4\right)\\ & & +{\left(5\cdot 5\right)}^{2}\left(n+4\right)+{\left(5\cdot 6\right)}^{2}\left(2n\right)+{\left(6\cdot 6\right)}^{2}\left(2n-3\right)\\ & = & 32+288+81n-162+450n+900+625n+2500+1800n\\ & & +2592n-3888\\ & = & 5548n-330 \end{array}$$ □

Now, we will calculate leap gourava indices for zigzag γ-graphyne nanoribbon.

**Theorem** **4.**
*Let ***Γ*** be a molecular graph of zigzag γ-graphyne nanoribbon and n>2. Then,*
*(i)* 

$LGO_{1}\left(\Gamma \right)=272n+162$

*(ii)* 

$LGO_{2}\left(\Gamma \right)=2064n+348$

*(iii)* 

$HLGO_{1}\left(\Gamma \right)=10478n+750$

*(iv)* 

$HLGO_{2}\left(\Gamma \right)=685264n-250392$

*(v)* 
*

$SLGO\left(\Gamma \right)=\left(\frac{1}{\sqrt{15}}+\frac{2}{\sqrt{23}}+\frac{1}{\sqrt{35}}+\frac{2}{\sqrt{41}}+\frac{2}{\sqrt{48}}\right)n+\left(\frac{2}{\sqrt{8}}+\frac{8}{\sqrt{11}}-\frac{2}{\sqrt{15}}+\frac{4}{\sqrt{23}}+\frac{4}{\sqrt{35}}-\frac{3}{\sqrt{48}}\right)$

*
*(vi)* 

$PLGO\left(\Gamma \right)=\left(\frac{1}{50}+\frac{1}{\sqrt{30}}+\frac{1}{\sqrt{54}}+\frac{2}{\sqrt{330}}+\frac{2}{\sqrt{432}}\right)n+\left(\frac{29}{50}+\frac{10}{\sqrt{30}}-\frac{2}{\sqrt{54}}-\frac{3}{\sqrt{432}}\right)$




**Proof.** (i)First Leap Gourava Index$$\begin{array}{ccc} LGO_{1}\left(\Gamma \right) & = & \sum_{\delta_{1}\delta_{2}\in E\left(\Gamma \right)}\left[\left(deg_{2}\left(\delta_{1}\right)+deg_{2}\left(\delta_{2}\right)\right)+\left(deg_{2}\left(\delta_{1}\right)\cdot deg_{2}\left(\delta_{2}\right)\right)\right]\\ & = & \left(4+4)(2)+(5+6)(8)+(6+9)(n-2)+(8+15)(2n+4\right)\\ & & +(10+25)(n+4)+(11+30)(2n)+(12+36)(2n-3)\\ & = & 16+88+15n-30+46n+92+35n+140+82n+96n-144\\ & = & 272n+162 \end{array}$$(ii)Second Leap Gourava Index$$\begin{array}{ccc} LGO_{2}\left(\Gamma \right) & = & \sum_{\delta_{1}\delta_{2}\in E\left(\Gamma \right)}\left[\left(deg_{2}\left(\delta_{1}\right)+deg_{2}\left(\delta_{2}\right)\right)\cdot \left(deg_{2}\left(\delta_{1}\right)\cdot deg_{2}\left(\delta_{2}\right)\right)\right]\\ & = & \left(4\cdot 4)(2)+(5\cdot 6)(8)+(6\cdot 9)(n-2)+(8\cdot 15)(2n+4)+(10\cdot 25)(n+4\right)\\ & & +(11\cdot 30)(2n)+(12\cdot 36)(2n-3)\\ & = & 32+240+54n-108+240n+480+250n+1000+660n+864n-1296\\ & = & 2064n+348 \end{array}$$(iii)First Hyper Leap Gourava Index$$\begin{array}{ccc} HLGO_{1}\left(\Gamma \right) & = & \sum_{\delta_{1}\delta_{2}\in E\left(\Gamma \right)}{\left[\left(deg_{2}\left(\delta_{1}\right)+deg_{2}\left(\delta_{2}\right)\right)+\left(deg_{2}\left(\delta_{1}\right)\cdot deg_{2}\left(\delta_{2}\right)\right)\right]}^{2}\\ & = & {\left(4+4\right)}^{2}\left(2\right)+{\left(5+6\right)}^{2}\left(8\right)+{\left(6+9\right)}^{2}\left(n-2\right)+{\left(8+15\right)}^{2}\left(2n+4\right)\\ & & +{\left(10+25\right)}^{2}\left(n+4\right)+{\left(11+30\right)}^{2}\left(2n\right)+{\left(12+36\right)}^{2}\left(2n-3\right)\\ & = & 1096+225n-450+1058n+2116+1225n+4900+3362n\\ & & +4608n-6912\\ & = & 10478n+750 \end{array}$$(iv)Second Hyper Leap Gourava Index$$\begin{array}{ccc} HLGO_{2}\left(\Gamma \right) & = & \sum_{\delta_{1}\delta_{2}\in E\left(\Gamma \right)}{\left[\left(deg_{2}\left(\delta_{1}\right)+deg_{2}\left(\delta_{2}\right)\right)\cdot \left(deg_{2}\left(\delta_{1}\right)\cdot deg_{2}\left(\delta_{2}\right)\right)\right]}^{2}\\ & = & {\left(4\cdot 4\right)}^{2}\left(2\right)+{\left(5\cdot 6\right)}^{2}\left(8\right)+{\left(6\cdot 9\right)}^{2}\left(n-2\right)+{\left(8\cdot 15\right)}^{2}\left(2n+4\right)\\ & & +{\left(10\cdot 25\right)}^{2}\left(n+4\right)+{\left(11\cdot 30\right)}^{2}\left(2n\right)+{\left(12\cdot 36\right)}^{2}\left(2n-3\right)\\ & = & 7712+2916n-5832+28800n+57600+62500n+250000+217800n\\ & & +373248n-559872\\ & = & 685264n-250392 \end{array}$$(v)Sum Connectivity Leap Gourava Index$$\begin{array}{ccc} SLGO(\Gamma ) & = & \sum_{\delta_{1}\delta_{2}\in E\left(\Gamma \right)}\frac{1}{\sqrt{\left[\left(deg_{2}\left(\delta_{1}\right)+deg_{2}\left(\delta_{2}\right)\right)+\left(deg_{2}\left(\delta_{1}\right)\cdot deg_{2}\left(\delta_{2}\right)\right)\right]}}\\ & = & \frac{1}{\sqrt{8}}\left(2\right)+\frac{1}{\sqrt{11}}\left(8\right)+\frac{1}{\sqrt{15}}\left(n-2\right)+\frac{1}{\sqrt{23}}\left(2n+4\right)+\frac{1}{\sqrt{35}}\left(n+4\right)\\ & & +\frac{1}{\sqrt{41}}\left(2n\right)+\frac{1}{\sqrt{48}}\left(2n-3\right)\\ & = & \left(\frac{1}{\sqrt{15}}+\frac{2}{\sqrt{23}}+\frac{1}{\sqrt{35}}+\frac{2}{\sqrt{41}}+\frac{2}{\sqrt{48}}\right)n\\ & & +\left(\frac{2}{\sqrt{8}}+\frac{8}{\sqrt{11}}-\frac{2}{\sqrt{15}}+\frac{4}{\sqrt{23}}+\frac{4}{\sqrt{35}}-\frac{3}{\sqrt{48}}\right) \end{array}$$(vi)Product Connectivity Leap Gourava Index$$\begin{array}{ccc} PLGO(\Gamma ) & = & \sum_{\delta_{1}\delta_{2}\in E\left(\Gamma \right)}\frac{1}{\sqrt{\left[\left(deg_{2}\left(\delta_{1}\right)+deg_{2}\left(\delta_{2}\right)\right)\cdot \left(deg_{2}\left(\delta_{1}\right)\cdot deg_{2}\left(\delta_{2}\right)\right)\right]}}\\ & = & \frac{1}{\sqrt{16}}\left(2\right)+\frac{1}{\sqrt{30}}\left(8\right)+\frac{1}{\sqrt{54}}\left(n-2\right)+\frac{1}{\sqrt{120}}\left(2n+4\right)+\frac{1}{\sqrt{250}}\left(n+4\right)\\ & & +\frac{1}{\sqrt{330}}\left(2n\right)+\frac{1}{\sqrt{432}}\left(2n-3\right)\\ & = & \left(\frac{1}{50}+\frac{1}{\sqrt{30}}+\frac{1}{\sqrt{54}}+\frac{2}{\sqrt{330}}+\frac{2}{\sqrt{432}}\right)n+\left(\frac{29}{50}+\frac{10}{\sqrt{30}}-\frac{2}{\sqrt{54}}-\frac{3}{\sqrt{432}}\right) \end{array}$$ □

## 4. Numerical Results and Discussion

In this section, we unveil a captivating array of numerical findings, delving into the profound realm of distance-degree-based topological indices applied to both γ-graphyne nanoribbon and zigzag γ-graphyne nanoribbon. The graphical representation of the leap Zagreb indices is shown in Figure 5, the hyper leap Zagreb indices are shown in Figure 6, the leap gourava indices are shown in Figure 7, and the hyper leap gourava indices are shown in Figure 8 for γ-graphyne nanoribbon.

In this study, we perform calculations to generate tables with numbers that represent different distance-degree-based indices. These indices include the leap Zagreb indices, hyper leap Zagreb indices, leap gourava indices, and hyper leap gourava indices. We vary the values of *m* and *n* to explore how these indices change. The results for γ-graphyne nanoribbon are organized in Table 1 and Table 2, and, for zigzag γ-graphyne nanoribbon, they are organized in Table 3 and Table 4. Furthermore, we create graphs in Figure 9 and Figure 10 for γ-graphyne nanoribbon and in Figure 11 and Figure 12 for zigzag γ-graphyne nanoribbon to visually analyze the patterns and trends of these topological indices. By selecting specific values of *m* and *n*, we can better understand how these indices behave and potentially discover interesting relationships between them.

A comparative analysis of the figures shows that the hyper leap Zagreb index and hyper leap gourava index both attain the highest values as compared to the others for both the γ-graphyne nanoribbon and zigzag γ-graphyne nanoribbon. Molecular topology and connectivity are linked to the characteristics of molecules that are associated with the hyper leap Zagreb index and hyper leap gourava index. The hyper leap Zagreb index is used to describe the degree of branching and connectivity in a molecular structure. It provides information about the complexity and branching pattern of the molecular structure. A molecule’s connection and branching pattern are considered while using the hyper leap gourava Index. It provides information about the distribution of branching within the molecule and the spatial arrangement of atoms.

## 5. Conclusions and Future Work

In conclusion, this study proves that topological indices can be used to pinpoint the physicochemical properties of chemical substances. The edge partition technique was employed in this study to obtain results related to distance-degree-based topological indices including leap Zagreb indices, hyper leap Zagreb indices, leap gourava indices, and hyper leap gourava indices. The γ-graphyne nanoribbon and zigzag γ-graphyne nanoribbon were the subject of investigation. By employing the edge partition technique, numerical results for these indices were obtained. The γ-graphyne nanoribbon and zigzag γ-graphyne nanoribbon both exhibit the largest values for the hyper leap Zagreb index and the hyper leap gourava index when compared to other indices, according to the calculated indices. The greatest hyper leap Zagreb index value suggests a highly linked and branched structure, which implies that the bonding patterns of the γ-graphyne nanoribbon and zigzag γ-graphyne nanoribbon are detailed and complex. For the γ-graphyne nanoribbon and zigzag γ-graphyne nanoribbon, the maximum value of the hyper leap gourava index suggests a diverse degree distribution paired with diverse vertex eccentricities, indicating a structurally rich and complex topology. These findings highlight the potential of topological indices and the edge partition technique in exploring and understanding the physicochemical properties of these structures. The outcomes of this research will contribute to the exploration of structure–property relationships in the studied materials. These findings will aid in understanding how the molecular structure of these materials influences their properties, thus paving the way for further investigations and applications in various fields.

In future research, we can calculate multiplicative distance-degree-based topological indices, entropy, M-polynomial indices to characterize the molecular structure of γ-graphyne nanoribbon and zigzag γ-graphyne nanoribbon. Conducting such an analysis will yield a further understanding of the properties and behavior of these materials, thereby potentially expanding their range of applications across various fields.

## Figures and Tables

**Figure 1 molecules-28-06597-f001:**
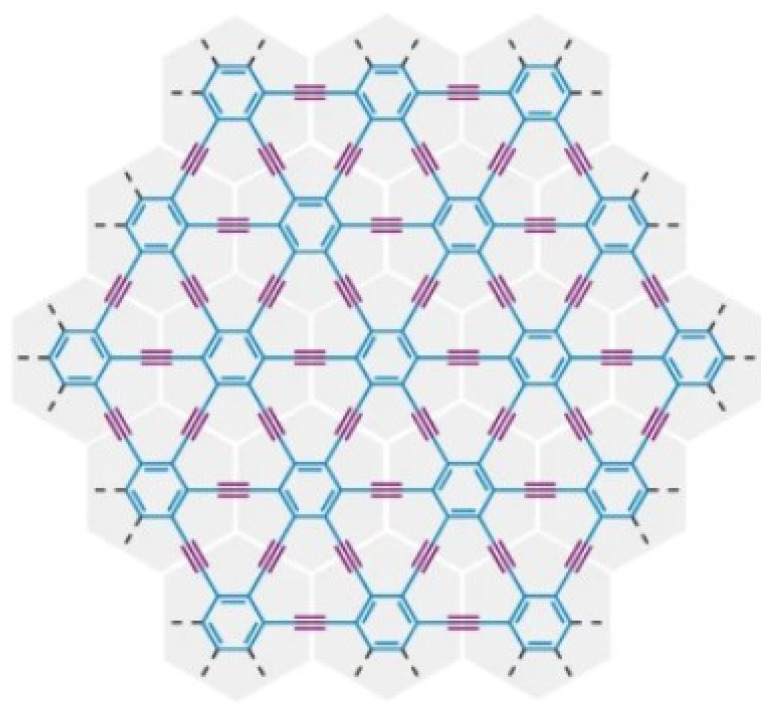
Structure of γ-graphyne.

**Figure 2 molecules-28-06597-f002:**
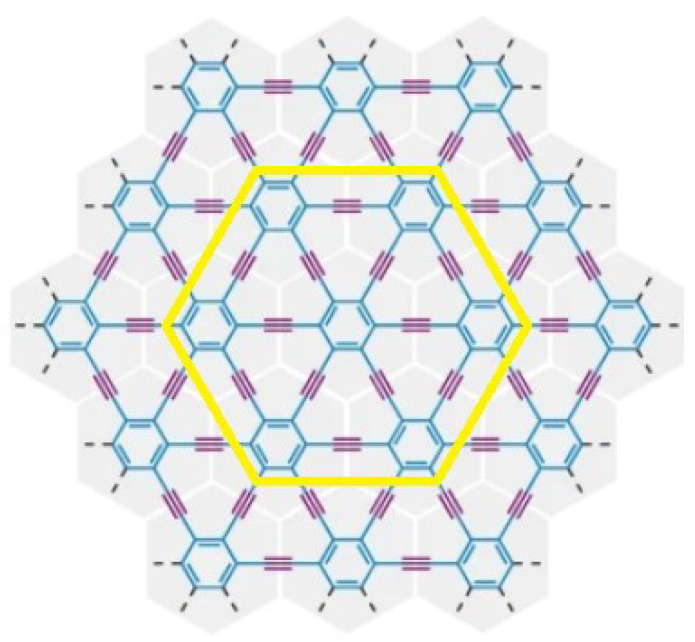
Structure of γ-graphyne.

**Figure 3 molecules-28-06597-f003:**
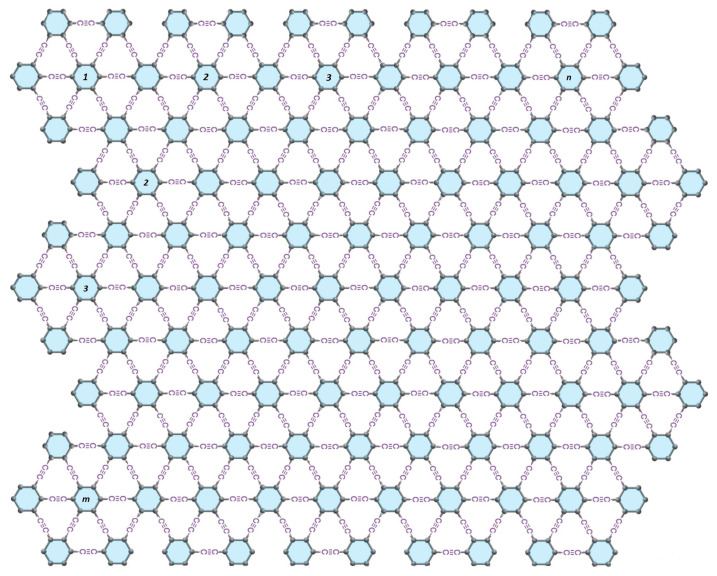
Structure of γ-graphyne nanoribbon with m,n≥1.

**Figure 4 molecules-28-06597-f004:**
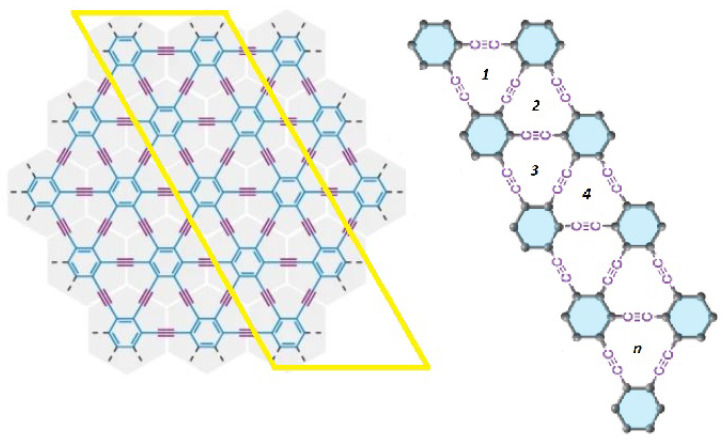
Zigzag γ-graphyne nanoribbon with n>2.

**Figure 5 molecules-28-06597-f005:**
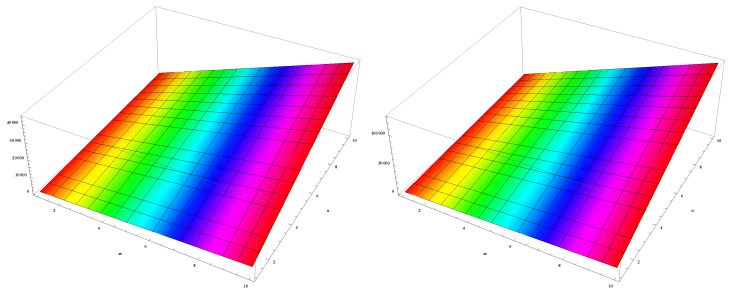
Graphical representation of leap Zagreb indices $LM_{1}\left(\Gamma \right)$ and $LM_{2}\left(\Gamma \right)$.

**Figure 6 molecules-28-06597-f006:**
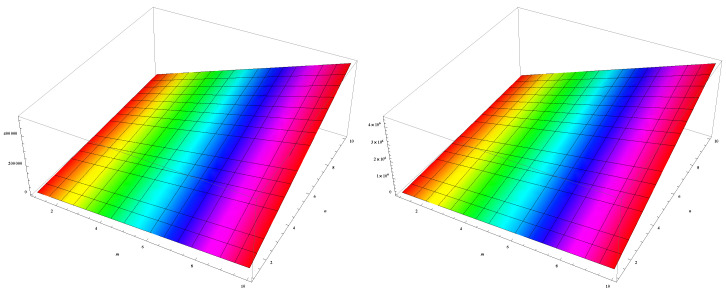
Graphical representation of hyper leap Zagreb indices $HLM_{1}\left(\Gamma \right)$ and $HLM_{2}\left(\Gamma \right)$.

**Figure 7 molecules-28-06597-f007:**
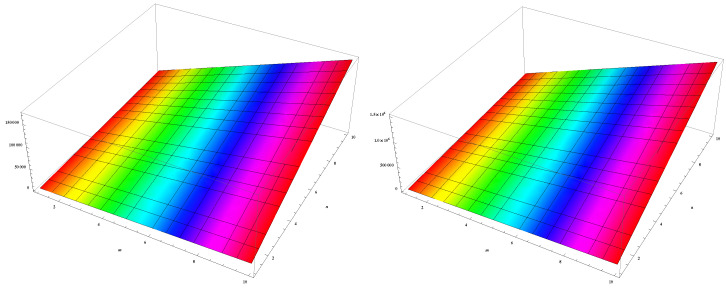
Graphical representation of leap gourava indices $LGO_{1}\left(\Gamma \right)$ and $LGO_{2}\left(\Gamma \right)$.

**Figure 8 molecules-28-06597-f008:**
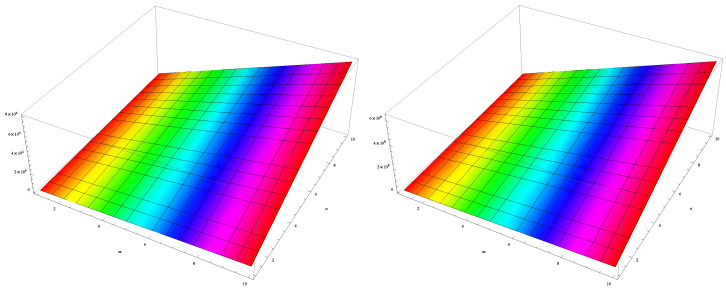
Graphical representation of hyper leap gourava indices $HLGO_{1}\left(\Gamma \right)$ and $HLGO_{2}\left(\Gamma \right)$.

**Figure 9 molecules-28-06597-f009:**
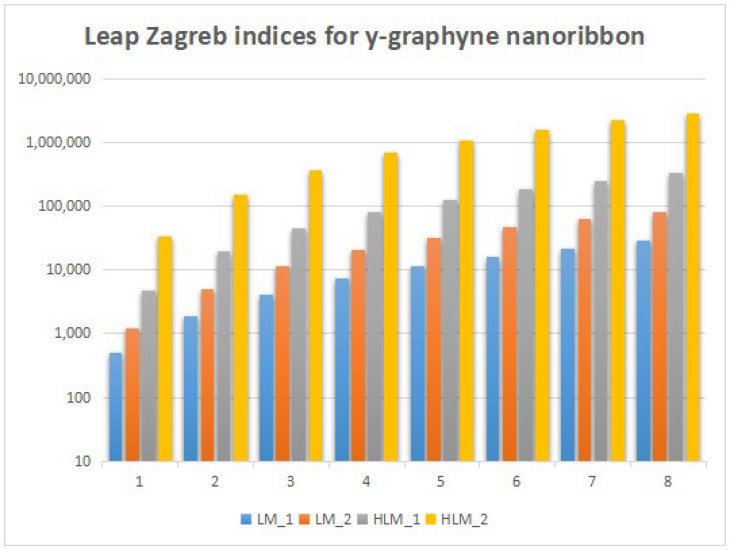
Logarithmic values of leap Zagreb indices for γ-graphyne nanoribbon.

**Figure 10 molecules-28-06597-f010:**
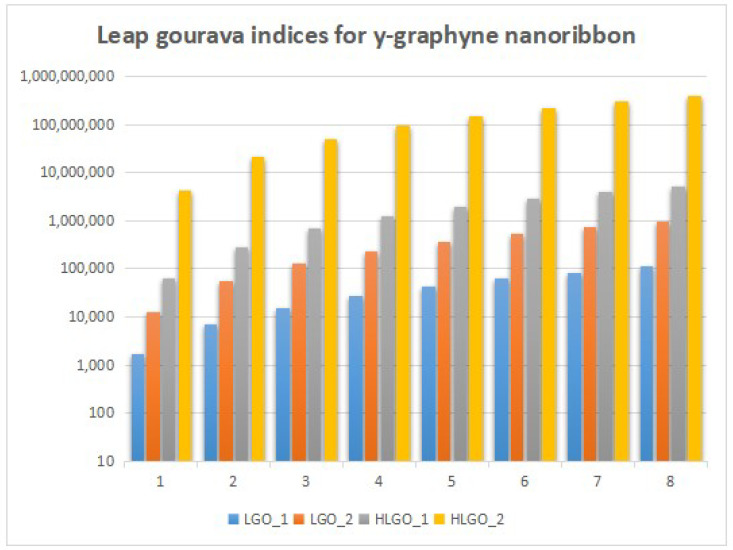
Logarithmic values of leap gourava indices for γ-graphyne nanoribbon.

**Figure 11 molecules-28-06597-f011:**
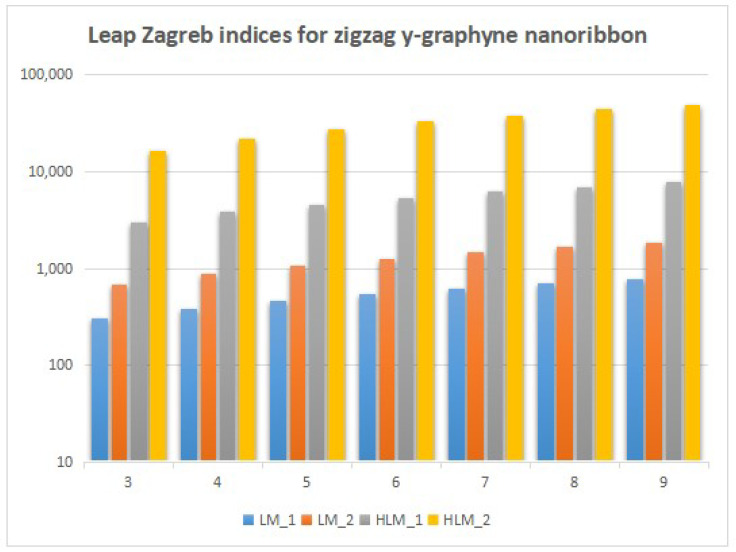
Logarithmic values of leap Zagreb indices for zigzag γ-graphyne nanoribbon.

**Figure 12 molecules-28-06597-f012:**
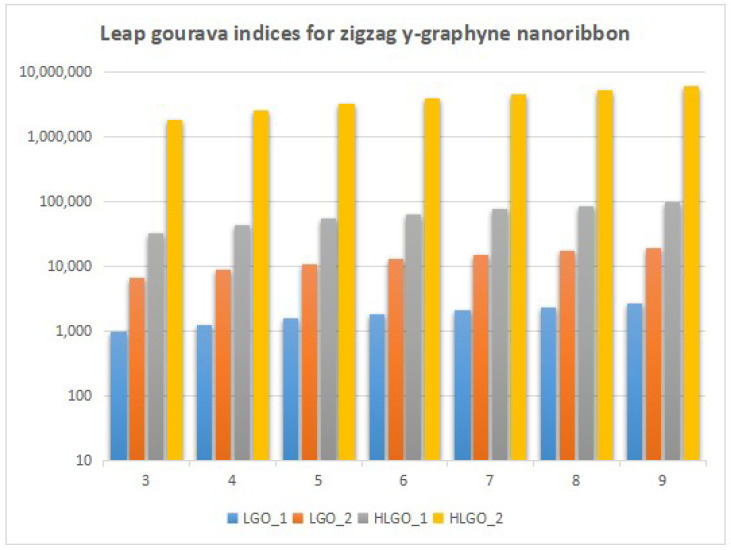
Logarithmic values of leap gourava indices for zigzag γ-graphyne nanoribbon.

**Table 1 molecules-28-06597-t001:** Numerical computations of leap Zagreb indices for γ-graphyne nanoribbon.

(m,n)	${\mathrm{LM}}_{1}\left(\Gamma \right)$	${\mathrm{LM}}_{2}\left(\Gamma \right)$	${\mathrm{HLM}}_{1}\left(\Gamma \right)$	${\mathrm{HLM}}_{2}\left(\Gamma \right)$
(1,1)	492	1194	4848	33,234
(2,2)	1868	4942	19,944	154,822
(3,3)	4108	11,282	45,408	369,722
(4,4)	7212	20,214	81,240	677,934
(5,5)	11,180	31,738	127,440	1,079,458
(6,6)	16,012	45,854	184,008	1,574,294
(7,7)	21,708	62,562	250,944	2,162,442
(8,8)	28,268	81,862	328,248	2,843,902

**Table 2 molecules-28-06597-t002:** Numerical computations of leap gourava indices for γ-graphyne nanoribbon.

(m,n)	${\mathrm{LGO}}_{1}\left(\Gamma \right)$	${\mathrm{LGO}}_{2}\left(\Gamma \right)$	${\mathrm{HLGO}}_{1}\left(\Gamma \right)$	${\mathrm{HLGO}}_{2}\left(\Gamma \right)$
(1,1)	1686	12,444	62,790	4,104,888
(2,2)	6810	54,868	284,502	20,650,216
(3,3)	15,390	128,396	671,922	50,632,472
(4,4)	27,426	233,028	1,225,230	94,051,656
(5,5)	42,918	368,764	1,944,426	150,907,768
(6,6)	61,866	535,604	2,829,510	221,200,808
(7,7)	84,270	733,548	3,880,482	304,930,776
(8,8)	110,130	962,596	5,097,342	402,092,672

**Table 3 molecules-28-06597-t003:** Numerical computations of leap Zagreb indices for zigzag γ-graphyne nanoribbon.

[n]	${\mathrm{LM}}_{1}\left(\Gamma \right)$	${\mathrm{LM}}_{2}\left(\Gamma \right)$	${\mathrm{HLM}}_{1}\left(\Gamma \right)$	${\mathrm{HLM}}_{2}\left(\Gamma \right)$
3	306	678	3022	16,314
4	384	874	3816	21,862
5	462	1070	4610	27,410
6	540	1266	5404	32,958
7	618	1462	6198	38,506
8	696	1658	6992	44,054
9	774	1854	7786	49,602

**Table 4 molecules-28-06597-t004:** Numerical computations of leap gourava indices for zigzag γ-graphyne nanoribbon.

[n]	${\mathrm{LGO}}_{1}\left(\Gamma \right)$	${\mathrm{LGO}}_{2}\left(\Gamma \right)$	${\mathrm{HLGO}}_{1}\left(\Gamma \right)$	${\mathrm{HLGO}}_{2}\left(\Gamma \right)$
3	978	6540	32,184	1,805,402
4	1250	8604	42,662	2,490,666
5	1522	10,668	53,140	3,175,930
6	1794	12,732	63,618	3,861,194
7	2066	14,796	74,096	4,546,458
8	2338	16,860	84,574	5,231,722
9	2610	18,924	95,052	5,916,986

## Data Availability

To support the findings of this study, no data were used.

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
