# Peer review of "Computational Studies on Diverse Characterizations of Molecular Descriptors for Graphyne Nanoribbon Structures"

_molecules, 2023, doi:10.3390/molecules28186597_

Round 1
Reviewer 1 Report
The manuscript “Computational Studies on Diverse Characterizations of Molecular Descriptors for Graphyne Nanoribbon Structures” is very interesting. As a very important two-dimensional carbon material, γ-graphyne is of great significance for scientific research and industrial production. Whether from the perspective of computational chemistry, materials chemistry, or applied mathematics,the exploration of molecular descriptors to characterize γ-graphyne is attractive. This manuscript, the authors compute distance-degree-based topological indices, the leap Zagreb indices, hyper leap Zagreb indices, leap gourava indices, and hyper leap gourava indices to characterize γ-graphyne and zigzag graphyne nanoribbon. The research method is reasonable and representative. Compared with Quantitative Structure-Property Relationship/Quantitative Structure-Activity Relationship (QSPR/QSAR) molecular analysis, the current method is more concise and clear. The author also introduces the basic principle and concept of this research method, which is necessary for the general reader. The author's presentation of the research results (including γ-Graphyne,Zigzag γ-Graphyne Nanoribbon,) is clear and gradual. The authors also investigated topological indices based on the degree of distance in the depth fields of γ-graphene and zigzag γ-graphene nanoribbons. The conclusions in the manuscript are consistent with the evidence and arguments presented. In addition, the references in the manuscript are appropriate. The figures and tables are suitable. The results of this manuscript are reliable and the conclusion is correct. Therefore, I recommend that the paper be accepted.
Author Response
We are very much thankful to the reviewer for especially sparing their precious time and forwarding useful comments.
Thanks!
Reviewer 2 Report
This manuscript presents the computation of a number of topological indices for gamma-graphyne. Unfortunately, it is not clear what new has been done in the description of the closed-shell expressions for topological indices LM1, LM2, HLM1, HLM2, LGO1, LGO2, HLGO1, HLGO2, SLGO, and PLGO for gamma-graphyne as functions of m and n as compared to the work of Hakeem, A.; Ullah, A.; Zaman, S.; T., “Computation of some important degree-based topological indices for γ-graphyne and Zigzag graphyne nanoribbon,” Molecular Physics 2023, vol. 121, e2211403. Apparently, the main result is the numerical assessments of these indexes for given m and n and plotting these as diagrams. If one has the ready-made expressions, one can immediately calculate these indexes for any pair m and n, but what sense does it make for a chemistry scientist? There have been told many vague words in the manuscript about the usefulness of these indexes for physical chemistry. Well, the Tables 1, 2 show these leap Zagreb indices and leap Gourava indices, calculated, by using the expressions proposed, apparently, by Hakeem and co-authors, and what of all of that numbers? What information about the thermodynamical or electric conductivity properties of gamma-graphyne one can see from the indexes of Tables 1, 2?
Briefly, the authors should clearly describe what new they have done as compared to the work of Hakeem and co-authors and for what purpose exactly these indexes will be useful in chemistry and in what way (which is of no less importance). If it will appear that they used the ready-made expressions from Hakeem and co-authors’s work, then the following minor remarks can be omitted, because, I will just reject the manuscript.
In general, the manuscript is poorly written in the sense of logic of narration, though, the style is not bad. The figures are in low quality, I advise to heighten the resolution. The explicit evaluation of polynomials in proofs to the theorems should be placed in the Supplementary section (especially, if these were already proven in the work of Hakeem and co-authors).
Minor remarks:
L. 45-49: I think that the lines 45-49 about graphene should precede the lines L40-45 about graphyne. Otherwise, it looks like the authors started the paragraph about graphyne, but in the middle of the paragraph they suddenly started discuss graphene, and then, starting from L. 49, they switched again to graphyne.
L. 45: “a single layer of carbon atoms” should be separated by commas.
L. 46-47: Graphene’s properties such as high electrical conductivity and exceptional strength should be referenced.
L. 53: “… novel form of carbon called graphene”? Maybe, the authors have meant “graphyne”?
L. 57-58: Please, provide the reference on the articles comparing the thermal conductivities of graphene and graphyne.
L. 70: Abbreviation QSPR/QSAR appears at the first time in line 70. It must be decrypted. QSPR – quantitive structure-property relations, QSAR – quantitive structure-activity relations.
L. 74. “In chemical …”. The “In” should not start from the capital letter.
L. 75: Refs. [9,10] should appear in L. 70, after introducing “QSPR/QSAR” at the first time.
L. 64-77: The whole paragraph is somewhat superfluous. For example, L. 73-74: “These indices serve as quantitative measures that aid in understanding various chemical properties and behaviors of molecules” – is practically the same as L. 66-67: “By examining topological indices associated with these molecular graphs, we can explore the physicochemical characteristics of the molecules.”
L. 78: The phrase “Mathematics, including polynomials and integers, plays a crucial role in …” seems a bit strange. Why the authors were in need to insert the remark “including polynomials and integers”? By this logic, one can list linear algebra or quaternion algebra, or whatnot. So, what was the point?
L. 85-86: The decryption of QSPR/QSAR should appear when it was first introduced in line 70.
L. 84-87: Please, provide the references to example of distance-degree-based indices in medication design and thermodynamics.
L. 86-92: This text is poorly structured and superfluous. Several times the authors mention about the applicability of topological indices to thermodynamics, several time they mention that these are useful in analyzing molecule (molecular analysis), and some other.
L. 117, 120: Not “somber”, but “sombor”.
L. 125: Please, decrypt “HVAP” as enthalpy of vaporization and “DHVAP” as standard enthalpy of vaporization.
L. 291: Vol. 121 was missed.
Style is good. Minor spell checking is advised. Logic of narration is poor.
Author Response
We are very much thankful to the reviewer for especially sparing their precious time and forwarding useful comments. In light of those, the revision of the manuscript has been amended and we sincerely believe that this has made the manuscript more interesting and informative.
Please see the attachment.
Thanks!

Round 2
Reviewer 2 Report
I recommend this manuscript for publishing. My main concern was clarified.
Author Response
Dear Respected Reviewer,
We are thankful for your precious comments, these comments make our article more effective and charming. Further, we are thankful for recommendation to publish our article in this journal.
Thanks and best regards,
Al authors,